# Perioperative Vascular Biomarker Profiling in Elective Surgery Patients Developing Postoperative Delirium: A Prospective Cohort Study

**DOI:** 10.3390/biomedicines9050553

**Published:** 2021-05-15

**Authors:** Jan Menzenbach, Stilla Frede, Janine Petras, Vera Guttenthaler, Andrea Kirfel, Claudia Neumann, Andreas Mayr, Maria Wittmann, Mark Coburn, Sven Klaschik, Tobias Hilbert

**Affiliations:** 1Department of Anesthesiology and Intensive Care Medicine, University Hospital Bonn, Venusberg-Campus 1, 53127 Bonn, Germany; jan.menzenbach@ukbonn.de (J.M.); stilla.frede@ukbonn.de (S.F.); s4japetr@uni-bonn.de (J.P.); vera.guttenthaler@ukbonn.de (V.G.); andrea.kirfel@ukbonn.de (A.K.); claudia.neumann@ukbonn.de (C.N.); maria.wittmann@ukbonn.de (M.W.); mark.coburn@ukbonn.de (M.C.); sven.klaschik@ukbonn.de (S.K.); 2Institute of Medical Biometrics, Informatics and Epidemiology (IMBIE), University Hospital Bonn, Venusberg-Campus 1, 53127 Bonn, Germany; amayr@uni-bonn.de

**Keywords:** postoperative delirium, biomarker, vascular inflammation, elective surgery, cardiac surgery, glycocalyx, CCL2, MCP-1

## Abstract

Background: Postoperative delirium (POD) ranks among the most common complications in surgical patients. Blood-based biomarkers might help identify the patient at risk. This study aimed to assess how serum biomarkers with specificity for vascular and endothelial function and for inflammation are altered, prior to or following surgery in patients who subsequently develop POD. Methods: This was a study on a subcohort of consecutively recruited elective non-cardiac as well as cardiac surgery patients (age > 60 years) of the single-center PROPDESC trial at a German tertiary care hospital. Serum was sampled prior to and following surgery, and the samples were subjected to bead-based multiplex analysis of 17 serum proteins (IL-3, IL-8, IL-10, Cripto, CCL2, RAGE, Resistin, ANGPT2, TIE2, Thrombomodulin, Syndecan-1, E-Selectin, VCAM-1, ICAM-1, CXCL5, NSE, and uPAR). Development of POD was assessed during the first five days after surgery, using the Confusion Assessment Method for ICU (CAM-ICU), the CAM, the 4-‘A’s test (4AT), and the Delirium Observation Scale (DOS). Patients were considered positive if POD was detected at least once during the visitation period by any of the applied methods. Non-parametric testing, as well as propensity score matching were used for statistical analysis. Results: A total of 118 patients were included in the final analysis; 69% underwent non-cardiac surgery, median overall patient age was 71 years, and 59% of patients were male. In the whole cohort, incidence of POD was 28%. The male gender was significantly associated with the development of POD (*p* = 0.0004), as well as a higher ASA status III (*p* = 0.04). Incidence of POD was furthermore significantly increased in cardiac surgery patients (*p* = 0.002). Surgery induced highly significant changes in serum levels of almost all biomarkers except uPAR. In preoperative serum samples, none of the analyzed parameters was significantly altered in subsequent POD patients. In postoperative samples, CCL2 was significantly increased by a factor of 1.75 in POD patients (*p* = 0.03), as compared to the no-POD cohort. Following propensity score matching, CCL2 remained the only biomarker that showed significant differences in postoperative values (*p* = 0.01). In cardiac surgery patients, postoperative CCL2 serum levels were more than 3.5 times higher than those following non-cardiac surgery (*p* < 0.0001). Moreover, after cardiac surgery, Syndecan-1 serum levels were significantly increased in POD patients, as compared to no-POD cardiac surgery patients (*p* = 0.04). Conclusions: In a mixed cohort of elective non-cardiac as well as cardiac surgery patients, preoperative serum biomarker profiling with specificity for vascular dysfunction and for systemic inflammation was not indicative of subsequent POD development. Surgery-induced systemic inflammation—as evidenced by the significant increase in CCL2 release—was associated with POD, particularly following cardiac surgery. In those patients, postoperative glycocalyx injury might furthermore contribute to POD development.

## 1. Introduction

Postoperative delirium (POD) ranks among the most common perioperative complications in surgical patients. Since the prevalence of POD is known to increase with patient age, and the average age of the general surgical patient population likewise increases, so does the incidence of POD. Comorbidities and the type of surgery do have a further impact, resulting in a statistical risk for POD ranging from 2.5% in the general surgical population up to 70% in patients requiring emergency femur fracture repair [1]. POD is characterized as cognitive impairment with an acute and fluctuating disturbance in awareness and attention, appearing in either hyper- or hypoactive manifestation or a combination of both [2]. Commonly occurring within the first five postoperative days, POD was shown to increase the length of stay (LOS) in the intensive care unit (ICU) and in hospital [3]. It is also associated with an up to 10 times increased 30-day mortality, as compared to individuals without POD [4]. Furthermore, the risk for long-term care dependency is likewise increased by up to three times [5]. Not least for these reasons, POD is associated with significantly increased health care costs and thus represents a substantial economic factor [3].

As strategies exist to reduce the risk for POD, especially in the vulnerable population, there is a particular need to identify those patients who are prone to develop POD. To this end, scores were developed and validated to determine the risk by including clinical parameters such as age, comorbidities, preoperative mental status, or details on surgery [6]. Blood-based biomarkers might provide additional information. It was shown that stress-induced mediators such as cortisol, C-reactive protein (CRP), or IL-6 might be associated with subsequent POD development [7]. The significant increase in the prevalence of delirium in septic patients provides further evidence for the role of systemic inflammation in delirogenesis [8].

Systemic inflammation and sepsis are furthermore associated with altered microvascular function, resulting in blood-brain barrier disruption, for example [9,10]. Meanwhile, biomarkers with specificity for vascular and endothelial injury, such as soluble adhesion molecules (Selectins and CAMs (Cellular Adhesion Molecules)), Angiopoietins, and soluble TIE2 receptor, or glycocalyx components (Syndecan-1) play a substantial role in estimating the prognosis of septic patients [11]. Moreover, clinical risk factors for vascular damage, including nicotine abuse, diabetes, or atherosclerosis as well as intraoperative vascular disturbances were shown to be associated with an increased risk for POD, underscoring the significance of impaired vascular and endothelial integrity for the pathogenesis of POD [12,13]. However, whether biomarkers with specificity for vascular function might be altered prior to or following surgery in patients that subsequently develop POD is not yet systemically investigated. Therefore, we assessed a panel of 17 biomarkers characteristic for vascular activation and permeability and for systemic inflammation in serum samples from elective adult patients across all relevant types of surgery. Analyses were performed on a subcohort of the single-center PROPDESC trial [14].

## 2. Materials and Methods

### 2.1. Study Design and Patient Population

The PRe-Operative Prediction of postoperative DElirium by appropriate SCreening (PROPDESC) trial is an investigator-initiated prospective monocentric observational study at the Department of Anesthesiology and Intensive Care Medicine, University Hospital Bonn. Details of the study design were previously described [14]. The trial was conducted in accordance with the Declaration of Helsinki and after approval by the institutional review board (IRB) of the University of Bonn (protocol number 255/17, date of approval 18 September 2017; Chairman—Professor K. Racké). The study was registered in the German Clinical Trials Register (protocol number DRKS00015715). Analyses of the herein presented study were performed on a subcohort of participants consecutively recruited between July and September 2019. Inpatients from various surgical specialties, including general, orthopedic, and trauma, cardiac, thoracic, vascular, ear–nose–throat, urologic, and plastic surgery were prospectively screened for eligibility to the study. Inclusion criteria were—elective surgery scheduled for a duration of at least 60 minutes, patient age >60 years, willingness and ability to provide written informed consent. Exclusion criteria were—emergency procedures, substantial language barriers, and a lack of patients’ compliance with the study protocol, which was determined by the respective physician. A checklist according to the STROBE statement can be found in Appendix A.

Recorded preoperative baseline characteristics included patient age, sex, body mass index (BMI), American Society of Anesthesiologists (ASA) Physical Status Classification System, surgical risk, surgical specialty, and preoperative routine laboratory values (hemoglobin (Hgb), HbA_1c_, leukocyte count, sodium, potassium, creatinine, CRP, total protein, high-sensitive cardiac troponin T (hs-TnT), and NT-proBNP). For surgical risk classification, the 5-level Johns Hopkins classification of intervention risk was transformed into a 3-level modified classification (low, intermediate, high risk (adapted from Glance et al. [15]).

### 2.2. Serum Sampling and Biomarker Profiling

All patients recruited to PROPDESC during the above-mentioned period were subjected to an additional serum biomarker profiling and included in the study. Ten milliliter of blood were drawn prior to and following surgery. The coagulated samples were centrifuged (3.000 rpm, 4 °C, 10 min), and serum aliquots were stored at −80 °C for subsequent analysis. In serum samples, the following 17 proteins were assessed using custom-made Luminex™ multiplex arrays (RnD Systems, Minneapolis, MN, USA):

Interleukin-3 (IL-3), IL-8, IL-10, Cripto, CC-chemokine Ligand 2 (CCL2), soluble Receptor for Advanced Glycation Endproducts (RAGE), Resistin, Angiopoietin-2 (ANGPT2), soluble Tyrosine Kinase with Immunoglobulin-like and EGF-like domains 2 (TIE2), Thrombomodulin (THBD), Syndecan-1 (SDC1), E-Selectin, soluble Vascular Cell Adhesion Protein 1 (VCAM-1), soluble Intercellular Adhesion Molecule 1 (ICAM-1), C-X-C Motif Chemokine 5 (CXCL5), Neuron-specific Enolase (NSE), and Urokinase Plasminogen Activator Surface Receptor (uPAR).

All analyses were performed according to the manufacturer’s protocol. Bead-based multiplex arrays such as the Luminex™ system are described in the work from Zhang et al. [16]. Arrays were analyzed using a MAGPIX™ reader (Luminex Corp., Austin, TX, USA). Results are given in pg/mL serum. Values below the assay’s lower detection limit (undetected values) were set to half the lower detection limit when the number of undetected values did not exceed 15% of all values of the respective protein in the whole cohort. When number of undetected values exceeded 15% of all values of the respective protein in the whole pre- as well as postoperative cohort, these analytes were totally excluded from further analysis. All experimental tests were performed in duplicates. The mean value was calculated from these results and used for further statistical analysis. All personnel performing the serum analyses were blinded for intra- and postoperative patient data.

### 2.3. Assessment of POD

POD assessment and postoperative data recording were implemented through regular patient rounds in the ICU and peripheral wards. Primary endpoint was development of POD within the first five days following surgery. POD assessments were performed every morning by trained study personnel on each of the first five days after surgery or after ending of sedation (RASS (Richmond Agitation and Sedation Scale) level ≥ −3), respectively. The Confusion Assessment Method for ICU (CAM-ICU) was used for ICU patients [17], while the Confusion Assessment Method (CAM) [18] and the 4-‘*A*’s test (4AT) [19] were used for the patients in the peripheral ward. In order not to miss any POD-positive patients due to spot examination, the Delirium Observation Scale (DOS) [20] was additionally used. A 4AT score of 4 points and a DOS score of 3 points onwards, respectively, was considered positive. Primary endpoint was considered to be achieved if POD was detected at least once during the 5-day visitation period by any of the applied assessment methods (CAM-ICU, CAM, 4AT, or DOS). If patients died during the 5-day visitation period, they were classified as delirious for subsequent analysis if they showed a positive test result prior to death. If not, they were excluded from later analysis.

Additionally recorded data included details on anesthesia and surgery (general or regional or combined anesthesia, duration of surgery, duration of mechanical ventilation, postoperative admission to ICU, and LOS in hospital).

### 2.4. Statistical and Bioinformatical Analysis

Structured patient data and results of POD assessment were entered into the electronic database “*REDCap*”, which is administered by the Institute for Medical Biometrics, Informatics and Epidemiology (IMBIE) of the University Hospital Bonn. All data including results from biomarker assessment were then transferred into MS Excel 2019 (Microsoft Corp., Redmond, CA, USA). Statistical analysis and visualization were performed using GraphPad PRISM 8.4.3 (La Jolla, CA, USA) and the statistical computing environment R 3.5.1 (Vienna, Austria).

All data are presented as percentage values or as median values with 25th and 75th percentile, respectively. To assess differences between groups (non-paired samples), non-parametric Mann-Whitney U test was used; in case of comparisons of pre- and postoperative measurements (paired samples), Wilcoxon rank-sum tests were applied. Binary variables were compared using Fisher’s exact test. *p* values < 0.05 (two-sided) were considered statistically significant.

For the analysis of serum samples, a nearest neighbor-based propensity score matching [21] was performed to generate a control group with similar baseline characteristics to the POD group. Variables considered for the logistic regression model to estimate the propensity score were surgical risk, CRP, leukocyte count, BMI, patient age, duration of surgery, and patient gender. For the comparison of the matched groups, the corresponding Wilcoxon rank-sum test for paired samples was applied.

The datasets generated and analyzed during the current study are available from the corresponding author on request.

## 3. Results

A total of 123 consecutive participants of the PROPDESC trial were prospectively recruited to this subcohort study. Five patients were excluded from later analysis as multiplex array analysis of postoperative serum samples failed, resulting in a total of 118 assessed patients with complete data, as well as serum sample sets. A total of 81 patients (69%) underwent non-cardiac surgery. All patients but one, who received conscious sedation, received general anesthesia. Overall median patient age was 71 years (66–78), and 70 patients (59%) were male. Three patients (2.5%) were classified ASA I, 35 patients (29.7%) ASA II, 70 patients (59.3%) ASA III, and 10 patients (8.5%) were classified as ASA IV. Risk of surgery was considered low in 11 cases (9.3%), intermediate in 47 cases (39.8%), while it was considered high in 60 cases (50.9%). Sixty-six patients (55%) were immediately admitted to ICU (including Intermediate Care Unit (IMC)), while the others were transferred to the peripheral ward from the recovery room. Details on further patient characteristics as well as on surgical procedures are given in Table 1.

Biomarker profiling was performed in pre- and in postoperative serum samples. In total, 236 serum samples were analyzed. For IL-3, IL-10, and Cripto, the number of undetected values exceeded 15% of all values of the respective protein in the whole pre- as well as postoperative cohort, therefore, these proteins were totally excluded from further analysis, resulting in 14 serum proteins being analyzed. As shown in Figure 1 (and in Appendix A), according to the Wilcoxon rank-sum test, surgery induced highly significant changes in serum levels of almost all biomarkers for vascular activation, permeability, and inflammation, with the exception of uPAR. While there was a considerable interindividual spread in pre- as well as postoperative markers, some of them were decreased in postoperative samples (E-Selectin, ICAM-1, THBD, TIE2, CXCL5), while others were increased following surgery, as compared to preoperative levels. Percentage decrease of median levels was most pronounced in TIE2 (reduction to 70.3%), and percentage increase was most pronounced in IL-8 (increase to 203%).

Development of POD was assessed during the first five days after surgery or after ending of sedation, respectively. In the whole cohort, incidence of POD was 28% (33 patients). While 18 patients were considered POD-positive for one day, POD remained for two days in 9 patients and for three or more days in 6 patients. A total of 20 patients developed POD on postoperative day 1 or 2 (early-onset), while 13 developed POD on day 3 to 5 (late-onset). According to the Fisher’s exact test, the male sex was significantly associated with the development of POD (28 male vs. 5 female patients, *p* = 0.0004), as was higher ASA status III (*p* = 0.04). In the POD subcohort, overall duration of mechanical ventilation was significantly increased (11.0 (4.6–26.1) vs. 5.1 (3.4–10.8) h), and immediate postoperative admission to ICU was more common (*p* = 0.008). In contrast, there was no difference in POD incidence between low to intermediate and high-risk surgery (*p* = 0.1) or concerning the preoperative routine laboratory parameters (*p* = 0.07). Duration of surgery was likewise not different between the POD and the no-POD group (249 (168–336) vs. 204 (132–278) min, *p* = 0.13). Details on patient characteristics as well as on surgical procedures in both subcohorts are given in Appendix A.

The results of the pre- and the postoperative multiplex biomarker assessment for the POD and the no-POD cohort, respectively, are shown in Figure 2 and in Appendix A. According to the Mann-Whitney test, in preoperative serum samples, none of the analyzed parameters was significantly altered in the POD patients, as compared to the no-POD cohort (Figure 2A). As for the whole cohort, there was a considerable interindividual spread. This also applied to the postoperative serum sample values. However, as shown in Figure 2B, values for CCL2 following surgery were significantly increased by a factor of 1.75 in POD patients (*p* = 0.03), as compared to the no-POD cohort.

Since some of the analyzed (particularly postoperative) biomarkers tended to be different in the POD cohort but without actually reaching statistical significance when being compared to the control cohort, a propensity score matching was performed in order to correct for the potential confounders and to increase statistical power. However, CCL2 remained the only biomarker showing significant differences in postoperative absolute (*p* = 0.01) as well as fold-change values (*p* = 0.03), while the preoperative samples showed no difference between the two subcohorts (Figure 3). Complete results of the matched-pairs serum analysis are given in Appendix A.

According to Fisher’s exact test, the incidence of POD was significantly increased in cardiac surgery patients, as compared to non-cardiac surgery (18 out of 37 (49%) vs. 15 out of 81 patients (19%), *p* = 0.002; Figure 4A). In cardiac surgery patients, CCL2 was the only one among all analyzed biomarkers that showed a highly significant difference in postoperative absolute (1117 (670.8–1749) vs. 315.9 (240.1–481.6) pg/mL), as well as fold-change increase (3.21 (2.51–6.17) vs. 0.93 (0.71–1.29)), irrespective of POD development, as compared to the non-cardiac surgery patients (*p* < 0.0001) (Figure 4B). Postoperative CCL2 serum levels were more than 3.5 times higher following cardiac surgery than following non-cardiac surgery. When the subcohort of cardiac surgery patients was furthermore divided according to subsequent POD development, only postoperative SDC1 serum levels were significantly increased in POD patients, as compared to no-POD cardiac surgery patients (3985 (3005–5871) vs. 2951 (2451–3959) pg/mL, *p* = 0.04) (Figure 4C).

## 4. Discussion

Vascular injury as well as clinically evident systemic inflammation were both associated with increased risk of POD [7,8,12]. Our herein presented results of the analysis of 17 serum biomarkers with specificity for vascular activation and permeability and for inflammation, revealed that almost all markers were significantly altered following surgery, as compared to the preoperative values. However, with the exception of CCL2, none of the other biomarkers analyzed was significantly different in pre- or in postoperative samples in patients that later developed POD, as compared to the no-POD subcohort. A paired analysis following propensity score matching confirmed the results. Incidence of POD was furthermore significantly increased following cardiac surgery, as compared to non-cardiac surgery patients. Among all biomarkers, CCL2 was the only one that showed a highly significant difference in postoperative increase, following cardiac surgery, as compared to the non-cardiac surgery patients, suggesting a critical role of proinflammatory activation in the pathogenesis of POD in this patient population. Furthermore, postoperative SDC1 serum levels were significantly increased in cardiac surgery patients who developed POD, as compared to those without subsequent POD.

POD is supposed to be mediated by neuroinflammation and might therefore be designated as an inflammatory “state of mind” [22]. Although still elusive, the underlying factors in the pathogenesis of POD appear to be numerous. Patient age and gender, preexisting cognitive decline, comorbidities including (among numerous others) alcohol abuse, diabetes, and hypertension, and emergency surgery were identified as predisposing factors that might not or at least might not be significantly influenced [1]. In contrast, precipitating factors such as perioperative pharmacological and anesthesiological management, extent and invasiveness of surgical measures, and preservation of fluid and temperature homeostasis can be actively controlled, thereby offering the chance to reduce the risk for POD development. The rationale of any preoperative screening instrument, whether score- or biomarker-based, is to identify those patients that might either benefit from further and thorough preoperative evaluation, from an adjustment in perioperative management, or from prolonged or more intense postoperative monitoring. Therefore, the first step in a successful reduction of the risk for POD is to identify the patient at risk.

In preoperative risk stratification, biomarkers are gaining more and more importance. Brain natriuretic peptide (BNP) or its N-terminal prohormone, for example, were shown to be validly associated with the incidence of myocardial injury, as well as with major adverse cardiac events after non-cardiac surgery [23,24]. Therefore, preoperative BNP assessment for perioperative cardiac risk stratification is recommended by recent national guidelines [25,26]. It was demonstrated that the predictive power of score-based screening tools for perioperative risk assessment was significantly improved when combined with biomarkers [27].

Myocardial injury, besides sepsis and bleeding, is still one of the most important determinants of postoperative mortality, therefore, the use of biomarkers with specificity for cardiac morbidity is most validated and established in preoperative risk assessment. However, the primary goal of perioperative medicine should not only be survival but rather a favorable treatment outcome in terms of physical, as well as mental performance, particularly avoiding cognitive decline to preserve a satisfying quality of life. The single-center PROPDESC trial was designed to identify score-based parameters that allow estimation of the patients’ individual risk for POD development from preoperative routine data in a cohort of elective surgery patients aged >60 years [14]. The results of the herein presented analysis were obtained from a large subcohort of consecutively enrolled PROPDESC patients who underwent additional biomarker profiling to identify potential candidate markers that might help improve score-based screening [27]. This subcohort comprised patients from relevant types of non-cardiac surgery as well as from cardiac surgery, and covered low- to high-risk procedures, thereby forming a representative sample from the population of elderly surgical patients at a German tertiary care hospital. The 14 assessed biomarkers were grouped into either vascular-specific or markers representing systemic inflammation. Surgery resulted in significant changes of postoperative serum levels in almost all biomarkers, as compared to preoperative levels. Increase in proinflammatory mediators following surgery was well described previously, and was among others, associated with the invasiveness of procedures [7,28,29]. In contrast, markers for vascular activation showed a more heterogenous kinetic. In line with results from others, some of them like E-Selectin, ICAM-1, or TIE2 were decreased in postoperative serum samples [29,30], while others (Syndecan-1, ANGPT2) were increased, indicating surgery-induced vascular and endothelial activation or injury [30,31].

Development of POD in our study was assessed using three different methodologies, each validated for the detection of hypo- or hyperactive or combined POD [17,18,19,20]. In our whole study cohort, 28% of patients were rated POD-positive during the first five days following surgery. Incidence of POD in cardiac surgery patients was significantly increased, as compared to non-cardiac surgery. These results were in line with previously reported epidemiological data [3,4,5,28]. When the whole study cohort was divided according to POD diagnosis, neither vascular activation nor inflammatory biomarkers significantly differed between the two groups prior to surgery, suggesting no association with subsequent POD development. Although only few authors assessed preoperative blood-based biomarkers in POD patients, this is at least partly in line with the results of others. In a recent meta-analysis, Liu et al. could show that preoperative serum levels of IL-8 and NSE were not altered in patients developing POD [7]. The same was demonstrated by Vasunilashorn et al. for IL-8 [28].

Given the impact of surgery on postoperative levels of almost all biomarkers, we also expected to see an association with subsequent POD development. However, only the serum levels of CCL2 following surgery were significantly increased in POD patients, as compared to the no-POD controls. This was a stable result of our analyses even after controlling for potential confounders by performing a propensity score matching, which additionally revealed a significant intergroup-difference in CCL2 fold-change levels. CCL2, also referred to as MCP-1 (Monocyte Chemoattractant Protein 1), is a chemokine primarily secreted by monocytes and macrophages in states of inflammation, exhibiting chemotactic properties on monocytes and subsets of granulocytes. Postoperative CCL2 release is directly correlated with the severity of surgical stress [32]. Systemic inflammation was shown to be associated with the risk for delirium [8]. In line with this, postoperative serum or plasma levels of CCL2 were found to be increased in elderly orthopedic patients developing POD, as compared to no-POD patients [33,34]. Increased CCL2 activity might furthermore induce acute neuroinflammation and thereby contribute to POD pathogenesis, as demonstrated by animal studies [22,35]. Appropriately, the upcoming INTUIT study will shed light on the role of neuroinflammation in POD pathogenesis, explicitly focusing on surgery-induced CCL2 release [36].

As cardiac surgery was demonstrated to be associated with the risk for POD [3,37], incidence was significantly increased in those patients in our study, as compared to non-cardiac surgery. Dividing the whole cohort into a cardiac and a non-cardiac surgery subcohort, this revealed that CCL2 serum levels were the only ones among all analyzed biomarkers of which the postoperative increase was markedly and significantly greater, following cardiac surgery, as compared to the other subcohort. While cardiac surgery induced a more than three-fold increase in non-cardiac surgery patients, postoperative CCL2 serum levels were further decreased rather than increased. This, on the one hand, again underlines the impact of surgical stress on systemic inflammation and CCL2 release, which furthermore was shown to explicitly follow cardiac surgery [32,38]. On the other hand, this might provide a mechanistic explanation for increased prevalence of POD in cardiac surgery patients, given the pathogenetic role of CCL2 for neuroinflammation and POD development [22,35].

When we focused our analysis on the subcohort of cardiac surgery patients, this furthermore revealed SDC1 as the only biomarker that showed a significant postoperative increase in POD patients, as compared to the no-POD group. SDC1 is a transmembrane heparan sulfate proteoglycan and as such a key component of the endothelial glycocalyx layer. Its serum levels are increased following surgery, suggesting endothelial damage induced by surgical trauma [30]. Particularly during cardiac surgery with cardiopulmonary bypass, vascular disturbances were associated with POD development [13], and markers of endothelial activation and damage including SCD1 were shown to be markedly increased [31,39]. Impairment of the glycocalyx with SDC1 release was furthermore demonstrated to be associated with delayed neuroinflammation, following cerebral hemorrhage [40]. A specific relation between SDC1 and POD development was not demonstrated so far but it seems likely, since vascular and endothelial injury were shown to be associated with the pathogenesis of delirium in septic patients [9,10,11]. Therefore, postoperative SDC1 release might serve as specific marker for increased risk for POD in cardiac surgery patients.

Our study has several strengths, but also specific limitations. First, unlike others, we assessed pre- as well as postoperative serum samples. This is justified by our aim to identify, on one hand, biomarkers associated with subsequent POD development that are altered already prior to surgery and thus might serve as ‘true’ predictors. On the other hand, with the assessment of perioperative change of biomarker serum levels (pre- vs. postoperative), we sought to shed further light on the pathogenesis of POD. However, our study lacks the inclusion of additional time-points of serum sampling, and therefore, later changes in biomarker profile remain undetected. Second, we drew a random sample of consecutive patients from the whole PROPDESC cohort in order to obtain representative and generalizable results for non-cardiac as well as cardiac surgery. The longitudinal design of the study results in little loss of follow-up. Patients aged < 60 years as well as neurosurgery patients were not included. This was done on one hand in order to increase overall POD prevalence in the whole study population (which is known to be associated with increasing age). On the other hand, we wanted to exclude the effect of intracranial surgical trauma on POD development. However, this might limit the transferability of our data to other patient populations. Consequently, our results should be confirmed in a larger multi-center approach. Third, the strength of our study was the use of multiple validated and complementary POD detection methods to ensure that not POD-positive cases were missed. We performed additional propensity score matching, which allowed for greater efficiency with the analysis of fewer samples in total, while controlling for potential confounders. We also examined 17 biomarkers with specificity for vascular and endothelial function and for inflammation, rather than assessing only one or a limited number of cytokines (as done in other studies). However, our selection missed some important candidate cytokines, which could also have served as internal methodological control, since, e.g., TNF-alpha was previously shown to be associated with POD [41]. Moreover, the use of multiplex arrays that allow for the measurement of a substantial number of biomarkers at once, while using only small serum sample sizes, could also have resulted in less accurate detection of some proteins over others. Therefore, when reproducing our results, the use of enzyme-linked immunosorbent assay as the ‘gold standard’ for cytokine assessment would be preferable. Last, as our data indicate, we did not succeeded in identifying preoperative candidate serum biomarkers that allow for the identification of the patient at risk, and this could also be seen as a shortcoming of our study.

## 5. Conclusions

In a mixed, representative cohort of elective non-cardiac as well as cardiac surgery patients, preoperative profiling including 17 serum biomarkers with specificity for vascular dysfunction and for systemic inflammation was not indicative of subsequent POD development. Surgery-induced systemic inflammation, evidenced by a significant increase in CCL2 release, was associated with POD, particularly following cardiac surgery. In those patients, postoperative glycocalyx injury might further contribute to POD development.

## Figures and Tables

**Figure 1 biomedicines-09-00553-f001:**
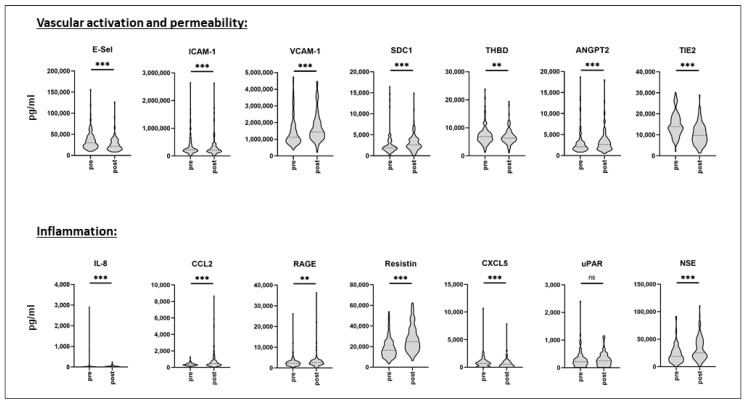
Pre- and postoperative serum biomarker profiling. Serum was sampled prior to and following surgery from a mixed cohort of consecutively recruited elective non-cardiac as well as cardiac surgery patients. Biomarker profiling was performed using the multiplex array technique. Data are given as median values (black dashed line) with 25th and 75th percentile (white dashed lines) and were compared using the Wilcoxon rank-sum test. *n* = 118. ns = not significant, ** *p* < 0.01, *** *p* < 0.005. E-Sel = E-Selectin, ICAM-1 = Intercellular Adhesion Molecule 1, VCAM-1 = Vascular Cell Adhesion Protein 1, SDC1 = Syndecan-1, THBD = Thrombomodulin, ANGPT2 = Angiopoietin-2, TIE2 = Tyrosine Kinase with Immunoglobulin-like and EGF-like domains 2, IL-8 = Interleukin-8, CCL2 = CC-chemokine Ligand 2, RAGE = Receptor for Advanced Glycation Endproducts, CXCL5 = C-X-C Motif Chemokine 5, uPAR = Urokinase Plasminogen Activator Surface Receptor, and NSE = Neuron.

**Figure 2 biomedicines-09-00553-f002:**
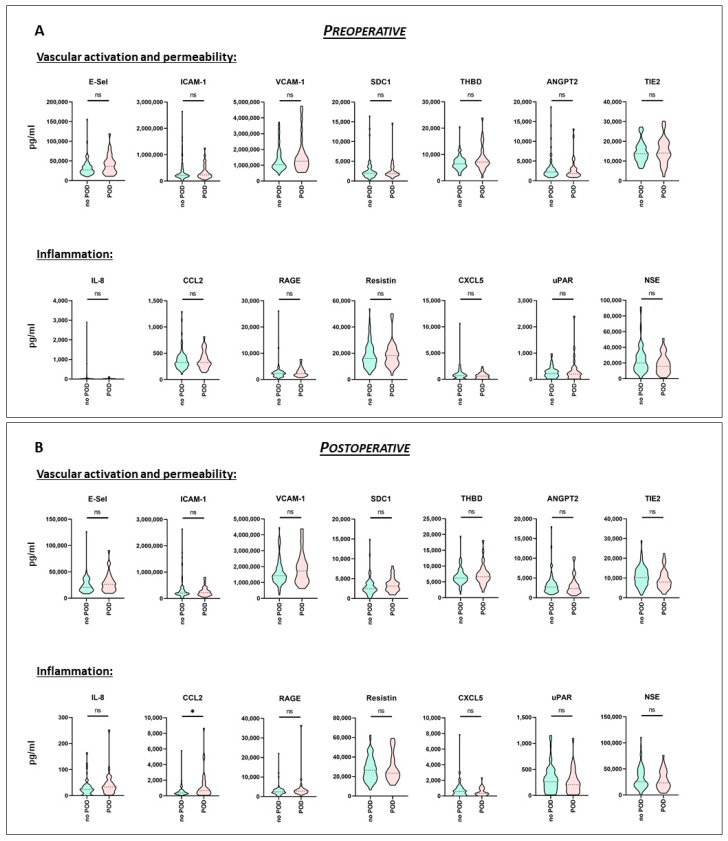
Pre- and postoperative serum biomarker profiling (no-POD and POD). Serum was sampled prior to (**A**) and following surgery (**B**) from a mixed cohort of consecutively recruited elective non-cardiac, as well as cardiac surgery patients. Biomarker profiling was performed using the multiplex array technique. The whole cohort was divided according to development of postoperative delirium (POD). Data are given as median values (black dashed line) with 25th and 75th percentile (white dashed lines) and were compared using the Mann-Whitney U test. no POD: *n* = 85, POD: *n* = 33. ns = not significant, * *p* < 0.05. E-Sel = E-Selectin, ICAM-1 = Intercellular Adhesion Molecule 1, VCAM-1 = Vascular Cell Adhesion Protein 1, SDC1 = Syndecan-1, THBD = Thrombomodulin, ANGPT2 = Angiopoietin-2, TIE2 = Tyrosine Kinase with Immunoglobulin-like and EGF-like domains 2, IL-8 = Interleukin-8, CCL2 = CC-chemokine Ligand 2, RAGE = Receptor for Advanced Glycation Endproducts, CXCL5 = C-X-C Motif Chemokine 5, uPAR = Urokinase Plasminogen Activator Surface Receptor, and NSE = Neuron-specific Enolase.

**Figure 3 biomedicines-09-00553-f003:**
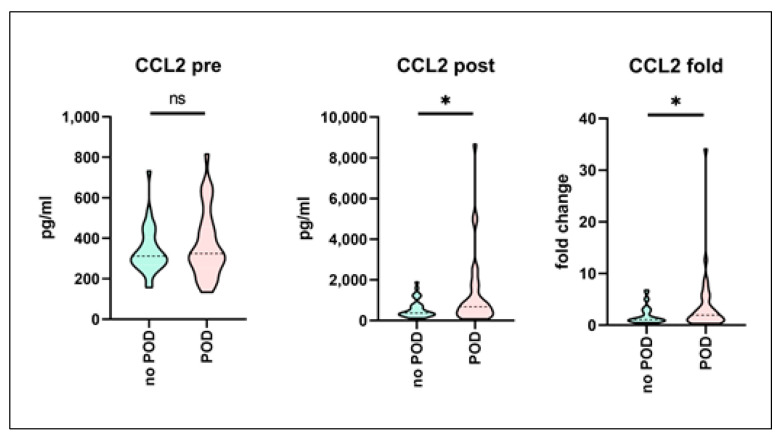
Pre- and postoperative serum biomarker profiling (no-POD and POD) following propensity score matching. Serum was sampled prior to and following surgery from a mixed cohort of consecutively recruited elective non-cardiac as well as cardiac surgery patients. Biomarker profiling was performed using the multiplex array technique. Development of postoperative delirium (POD) was assessed, and POD-positive patients were matched with respect to no-POD controls using propensity score matching. Figure shows results for the CCL2 (CC-chemokine Ligand 2) serum levels. Data are given as median values (black dashed line) with 25th and 75th percentile (white dashed lines) and were compared using Wilcoxon rank-sum test. no POD: *n* = 33, POD: *n* = 33. ns = not significant, * *p* <0.05.

**Figure 4 biomedicines-09-00553-f004:**
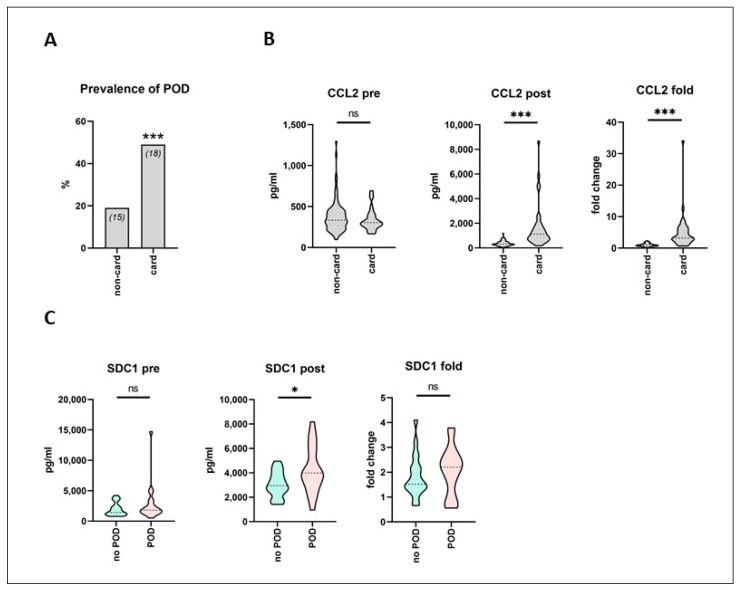
Pre- and postoperative serum biomarker profiling in cardiac and non-cardiac surgery patients. Serum was sampled prior to and following surgery from a mixed cohort of consecutively recruited elective non-cardiac as well as cardiac surgery patients. Biomarker profiling was performed using multiplex array technique. Development of postoperative delirium (POD) was assessed. (**A**) Prevalence of POD was significantly increased in cardiac surgery patients, as compared to non-cardiac surgery (18 out of 37 (49%) vs. 15 out of 81 patients (19%)). Fisher’s exact test. *** *p* < 0.005. (**B**) In cardiac surgery patients, CCL2 (CC-chemokine Ligand 2) showed a significant difference in postoperative absolute as well as fold-change increase, irrespective of POD development, as compared to the non-cardiac surgery patients. non-cardiac: *n* = 81, cardiac: *n* = 37. (**C**) In cardiac surgery patients, postoperative SDC1 (Syndecan-1) serum levels were significantly increased in patients that subsequently developed POD, as compared to the no-POD cardiac surgery patients. no POD: n = 19, POD: *n* = 18. (**B**,**C**) Data are given as median values (black dashed line) with 25th and 75th percentile (white dashed lines) and were compared using the Mann-Whitney U test. ns = not significant, * *p* < 0.05, *** *p* < 0.005.

**Table 1 biomedicines-09-00553-t001:** Patient and procedural details.

Parameter	Median (25th–75th Percentile)
*Patient details:*	
*n*	118
Age (years)	71 (66–78)
Male gender (*n* [%])	70 (59)
Body mass index (kg/m^2^)	27.3 (24.4–30.2)
ASA status:	
I (*n* [%])	3 (2.5)
II (*n* [%])	35 (29.7)
III (*n* [%])	70 (59.3)
IV (*n* [%])	10 (8.5)
*Preoperative routine laboratory values:*	
Hemoglobin (g/dL)	13.4 (12.2–14.5)
HbA_1C_ (%)	5.6 (5.4–6.2)
Leukocyte count (G/L)	6.9 (5.9–8.8)
Sodium (mmol/L)	140 (138–142)
Potassium (mmol/L)	4.4 (4.1–4.7)
Creatinine (mg/dL)	0.9 (0.78–1.05)
C-reactive protein (mg/L)	3.1 (1.1–8.8)
Total protein (g/L)	69 (65–73)
High-sensitive cardiac troponin T (ng/L)	12.9 (8.1–20.3)
NT-proBNP (pg/mL)	235 (114–796)
*Procedural and anesthesia details:*	
Surgical risk:	
Low (*n* [%])	11 (9.3)
Intermediate (*n* [%])	47 (39.8)
High (*n* [%])	60 (50.9)
Surgical specialty:	
General (*n* [%])	18 (15)
Orthopedic and trauma (*n* [%])	35 (30)
Cardiac (*n* [%])	37 (31)
Thoracic (*n* [%])	2 (2)
Vascular (*n* [%])	4 (3)
Ear-nose-throat (*n* [%])	11 (9)
Urologic (*n* [%])	9 (8)
Plastic (*n* [%])	2 (2)
Placement of epidural catheter (*n* [%])	6 (5)
Duration of surgery (min)	219 (136–298)
Duration of mechanical ventilation (h)	6.0 (3.6–14.0)
Postoperative admission to ICU (*n* [%])	66 (55)
Length of hospital stay (days)	13 (10–23)

Data are given as percentage values or as median values with 25th and 75th percentile, respectively. ASA = American Society of Anesthesiologists, NT-proBNP = N-terminal prohormone of brain natriuretic peptide, and ICU = Intensive Care Unit.

## Data Availability

All datasets generated and analyzed during the current study are available from the corresponding author on request.

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
