# Peer review of "Perioperative Vascular Biomarker Profiling in Elective Surgery Patients Developing Postoperative Delirium: A Prospective Cohort Study"

_biomedicines, 2021, doi:10.3390/biomedicines9050553_

Round 1
Reviewer 1 Report
Please, add a STROBE checklist.
- ABSTRACT.
- Abstract appropriately summarize the manuscript.
- There aren´t discrepancies between the Abstract and the remainder of the manuscript.
- The Abstract can be understood without reading the manuscript.
- INTRODUCTION.
- The Introduction is concise.
- The purpose of the study is well defined.
- The authors provide a rationale for performing the study based on a review of the medical literature with an appropriate length.
- This manuscript is Original Research, with a well-defined hypothesis.
- MATERIAL AND METHODS
- Please, clarify the study Design.
- Please, justify the choice of their study design.
- Test methods are well stated.
- RESULTS
- The results are cleary explained.
- Please, try to do more parallel the order of presentation of the results and the order of presentation of the methods.
- The results are reasonable and expected.
- There aren´t results introduced that are not preceded by an appropriate discussion in the Methods section.
- DISCUSSION
- The discussion is concise.
- Their research question was answered.
- Authors’ conclusions are justified by the results found in the study.
- Please, note limitations of the study.
- FIGURES
- Figures are appropriate and they are appropriately labeled.
- Adequately show the important results.
- TABLES
- Appropriately describe the results.
- REFERENCES
- The reference list follows the format for the journal.
Author Response
Dear Prof. Mousa,
dear Dr. Jeon,
dear Reviewers,
dear Editors,
we thank you for giving us the opportunity to submit a revised version of our manuscript. We have carefully studied the Reviewers’ comments and gratefully acknowledge the helpful suggestions they made to further improve our report. In order to meet their demands, we have made comprehensive modifications within the text, which are further explained in a detailed point-to-point reply. All changes made to the text are highlighted.
Reviewer #1:
Please, add a STROBE checklist.
We thank the Reviewer for this valuable comment. A STROBE checklist has now been uploaded as Supplementary Material.
MATERIAL AND METHODS
- Please, clarify the study Design.
- Please, justify the choice of their study design.
In order to meet the Reviewers’ demands, we now have added a paragraph to the Discussion section, comprehensively dealing with the strengths and limitations of our study. This paragraph also clarifies and justifies the design we chose for our study. We hope, these additional information satisfy the Reviewer’s demands.
Results
- Please, try to do more parallel the order of presentation of the results and the order of presentation of the methods.
To make this clearer to the reader, we now have swapped the order of the presentation of POD assessment and biomarker analysis in the Methods section. This now fits better to the way these data are presented in the Results section.
Discussion
Please, note limitations of the study.
We greatly thank you for this useful comment. In order to meet the Reviewers’ demands, we now have added a paragraph to the Discussion section, comprehensively dealing with the strengths and limitations of our study. Please do not hesitate to contact us again if you think that important information is still missing.
We are sure that these additional clarifications address all of your concerns sufficiently and that you agree that the changes improved our manuscript and helped to clarify the report. Please do not hesitate to contact us if you should have any more questions. We are very looking forward to your decision.
Thank you very much!
Sincerely,
PD Dr. med. Tobias Hilbert, MD, D.E.S.A.

Reviewer 2 Report
Brief Summary: This study is about occurrence of delirium in adult >60 years patients undergone an elective cardiac and non-cardiac surgery. Author’s aim was to investigate alterations of serum biomarkers with specificity for vascular and endothelial function and for inflammation (17 protein multiplex array) prior to or following the surgery in patients subsequently developing post-operative delirium (POD). Patients recruited is a sub-cohort of participants of The PRe-Operative Prediction of postoperative DElirium by appropriate SCreening (PROPDESC) trial from University Hospital Bonn (German Clinical Trials Register (“Deutsches Register Klinischer Studien”. protocol number DRKS00015715). Incidence of POD was 28% (33/118) overall, significantly increased in cardiac surgery patients .Authors presented that markers, although significantly altered following surgery compared to pre-op values, were not significantly differed in patients developed POD compared to non-POD, with exception of CCL2 (propensity matching score confirmation). Authors conclude that, serum biomarkers studied were not indicative of POD development, and only surgery-induced systemic inflammation, evidenced by significant increase in CCL2 release, was associated with POD, particularly following cardiac surgery. In those patients, postoperative glycocalyx injury may furthermore contribute to POD development.
Broad comments: This study deals with an extremely important issue occurring in patients following a surgery, considering that delirium affects short- and long-term outcome in elective surgical patients. The article is well-organized, well-written and easy to follow. Separate sections, Introduction, Methodology etc, are well-developed and synthesizing the literature, Delirium assessment tools used are validated and the stuff trained. Again, well-explained sound methodology and clear message, though negative results with no significant associations.
Specific comments: In detail:
Abstract: Explanatory, aim, results and conclusion clear.
Introduction: Easy to follow, short, clear aim of the study.
Methodology: The methodology is clearly explained and easy for another researcher to repeat it. Methodology is appropriately structured.
Results: All results are well-written and Results found in the study are almost negative for serum biomarkers to predict patients that will develop POD, except CCL2. and there is a general confusing sense of all these reported results.
Table1 has a typo (Sodium 14, instead of 140)
Discussion: The first paragraph gathers all information nicely of what this study found and added to the literature, although most of the results were negative. Authors critically and in depth discuss their results.
Limitations of their study are not clearly addressed.
Authors also state that the most important is to identify the patient at risk of developing POD, something that their study did not identify after combining clinical elements and protein (biomarkers) alteration pre- and post-operation.
Conclusions: Clear
Thank you for the privilege of reviewing your work.
Author Response
Dear Prof. Mousa,
dear Dr. Jeon,
dear Reviewers,
dear Editors,
we thank you for giving us the opportunity to submit a revised version of our manuscript. We have carefully studied the Reviewers’ comments and gratefully acknowledge the helpful suggestions they made to further improve our report. In order to meet their demands, we have made comprehensive modifications within the text, which are further explained in a detailed point-to-point reply. All changes made to the text are highlighted.
Reviewer #2:
Table1 has a typo (Sodium 14, instead of 140)
We thank the Reviewer for very carefully reading our manuscript. We have corrected this typo.
Limitations of their study are not clearly addressed.
Again, we greatly appreciate this useful comment. As mentioned above, we now have added a paragraph to the Discussion section that comprehensively deals with the strengths as well as limitations of our study. We hope this meets your demands.
We are sure that these additional clarifications address all of your concerns sufficiently and that you agree that the changes improved our manuscript and helped to clarify the report. Please do not hesitate to contact us if you should have any more questions. We are very looking forward to your decision.
Thank you very much!
Sincerely,
PD Dr. med. Tobias Hilbert, MD, D.E.S.A.
